# A network pharmacology approach to decipher the total flavonoid extract of Dracocephalum Moldavica L. in the treatment of cerebral ischemia- reperfusion injury

Xu Hu[1]☺, Yideresi Mola[2]☺, Wen-ling Su[1], Yue Wang[1], Rui-fang Zheng[1,3]*, Jian-guo Xing[1]*

1 Xinjiang Institute of Materia Medica, Xinjiang Key Laboratory of Uygur Medical Research, Urumqi, Xinjiang, China, 2 College of traditional Chinese medicine, Xinjiang Medical University, Urumqi, Xinjiang, China, 3 Department of Clinical Pharmacy, School of Preclinical Medicine and Clinical Pharmacy, China Pharmaceutical University, Nanjing, Jiangsu, China

☺ These authors contributed equally to this work.
* 872780352@qq.com (RFZ); xjguodd@163.com (JGX)

**Data Availability Statement:** All relevant data are within the paper and its Supporting Information files.

## Abstract

### Background and objective

Cerebral ischemia-reperfusion injury (CIRI) is a major injury that seriously endangers human health and is characterized by high mortality and high disability. The total flavonoid extract of *Dracocephalum moldavica* L.(TFDM) in the treatment of CIRI has been proved by clinical practice. But the mechanism for the treatment of CIRI by TFDM has not been systematically revealed.

### Study design and methods

The active compounds contained in TFDM were screened by literature mining and pharmacokinetic parameters, and the targets related to CIRI were collected by searching Drugbank, Genecards and OMIM databases. Cytoscape software was used to construct the protein interaction network of TFDM for the prevention and treatment of CIRI. Geneontology and signal pathway enrichment were analyzed. The key target pathway network of TFDM compounds was constructed and verified by pharmacological experiment *in vitro*.

### Results

21 active components were screened, 158 potential drug targets for the prevention and treatment of CIRI were obtained, 53 main targets were further screened in the protein-protein interaction network, and 106 signal pathways, 76 biological processes, 26 cell components and 50 molecular functions were enriched (P<0.05). Through the compound-target-pathway network, the key compounds that play a role in the prevention and treatment of CIRI, such as acacetin, apigenin and other flavonoids, as well as the corresponding key targets and key signal pathways, such as AKT1, SRC and EGFR were obtained. TFDM significantly decreased LDH, MDA levels and increased the NO activity levels in CIRI. Further

**Funding:** Funding: 1)Xu Hu, this work was funded by the Scientific Research Business Expenses of Xinjiang Uygur Autonomous Region (ky2022143). 2)Rui-fang Zheng, this work was funded by the Scientific Research Business Expenses of Xinjiang Uygur Autonomous Region (ky2022140) and National Natural Science Foundation of China (82204767). 3)Jian-guo Xing, this work was funded by the Xinjiang Uygur Autonomous Region Key Laboratory Open Project (2020D04020), National Natural Science Foundation of China (82260845,U1803281), Special project of the central government to guide local science and technology development (ZYYD2022A02). The funders had no role in study design, data collection and analysis, decision to publish, or preparation of the manuscript.

**Competing interests:** The authors have declared that no competing interests exist.

**Abbreviations:** CIRI, cerebral ischemia-reperfusion injury; TFDM, total flavonoid extract of *Dracocephalum moldavica* L.; rtPA, recombinant tissue plasminogen activator; TCM, Traditional Chinese Medicine; DML, *Dracocephalum Moldavica* L.; EPDM, effective parts of dracocephalum moldavica; PPI, protein-protein interaction; GO, Gene ontology; KEGG, Kyoto Encyclopedia of Genes and Genomes; MF, molecular function; BP, biological process; CC, cellular components; HRP, horseradish peroxidase; ANOVA, analysis of variance; SD, standard deviation; RIPA, radio immuno precipitation assay; PMSF, phenyl methyl sulfonyl fluoride; LDH, lactate dehydrogenase; MDA, malondialdehyde; NO, nitric oxide.

studies have shown that TFDM increases the number of SRC proteins, and TFDM also increases p-AKT/ AKT. Molecular docking results showed that acacetin-7-O (- 6''-acetyl) -glucopyranoside, acacetin7-O-β-D-glucopyranoside, apigenin-7-O-β-D-galactoside respectively had good affinity for SRC protein. Acacetin-7-O (- 6''-acetyl) -glucopyranoside, acacetin-7-O-β-D-glucuronide, acacetin7-O-β-D-glucopyranoside had good affinity for AKT1 protein, respectively.

## Conclusion

Our research showed that TFDM had the characteristics of multi-component, multi-target and multi-channel in the treatment of CIRI. The potential mechanism may be associated with the following signaling pathways:1) the signaling pathways of VEGF/SRC, which promote angiogenesis, 2) the signaling pathways of PI3K/AKT, which inhibit apoptosis, and 3) acacetin-7-O (- 6''-acetyl) -glucopyranoside is expected to be used as a candidate monomer component for natural drugs for further development.

## 1. Introduction

Stroke is one of the three major diseases affecting human health. At present, the domestic and foreign guidelines recommend the use of intravenous recombinant tissue plasminogen activator (rtPA) thrombolysis for recanalization in the acute phase. Thrombolytic therapy has a strict treatment time window. Thrombolytic therapy beyond the time window can achieve certain therapeutic effects, but reperfusion will cause further damage to the brain tissue and expand the degree of brain injury caused by cerebral ischemia, which becomes cerebral ischemia-reperfusion injury (CIRI) [1]. Therefore, how to protect the ischemic penumbra, prevent or reduce CIRI, and improve the prognosis of stroke has become a hot topic and focus of many studies. CIRI is a dynamically changing process involving multiple links, including oxidative stress, inflammatory response, energy metabolism disorder, excitatory amino acids toxicity, apoptosis, and mitochondrial dysfunction. However, a single compound cannot exert the therapeutic effect on multiple links, and Chinese medicine with multi-component and multi-target treatment characteristics can achieve this goal. Based on modern biological research, it is considered that the cascade reaction after ischemia is the biological basis of "toxin damaging brain collaterals" [2]. Multi-components in Chinese medicine can interfere with multiple links of cerebral ischemia to block the cascade reaction of brain injury, which has a good application prospect in the treatment and prevention of cerebral ischemic diseases.

In recent years, network pharmacology has prompted traditional Chinese medicine to use modern biochemical measurements and tools to improve or increase diagnostic descriptions, and the use of western concepts based on biochemistry, pathways or regulatory processes to explain TCM theory [3]. The combination of systems biology and pharmacology provides a new network / path analysis method for the treatment of complex diseases by traditional Chinese medicine, that is, pharmacokinetic evaluation and targeted prediction [4, 5]. As a comprehensive method for systematically investigating and interpreting TCM and its compound molecular mechanism, TCM network pharmacology represented by a complex biological system, breaking the traditional analytical chemistry and pharmacology techniques, effectively establishing a "compound-protein/gene-disease" network and revealing the principle of regulating small molecules in a high-throughput manner. Therefore, TCM network pharmacology has become a very effective drug combination analytical tool, which providing new ideas and means for the study of Chinese medicines with multi-component and multi-target synergy.

*Dracocephalum Moldavica* L. (DML) is a traditional Uighur medicine with the Uyghur name "BadiRanjibuya", which is used in Xinjiang and included in the Uighur classical medical book "Aricanon". *Dracocephalum moldavica* L. is a plant of Cymbidium in Labiatae, and the whole herb has a medicinal history of hundreds of years. It has the effects of benefiting heart and protecting brain, dispelling wind and heat, and opening occlusion. The Yixin Badirange-buya Granules, with its single component, is clinically used for the treatment of fatigue, insomnia, upset asthma, and neurasthenia [6]. The effective parts of dracocephalum moldavica (EPDM) is an effective part extracted and separated from dracocephalum moldavica, and the content of EPDM accounts for 53.06% of the total extract, and the main active components are flavonoid compounds such as naringin, Robinin -7-O-β-D- glucuronide, luteolin -7-O-β-D- glucuronide, geranioin -7-O-β-D- glucuronide and apigenin -7-O-β-D- glucuronide [7]. A study has shown that EPDM can effectively inhibit the inflammatory cascade after cerebral ischemia-reperfusion injury and protect the brain tissue [8].

In the present study, we used computational tools and resources to investigate the pharmacological network on CIRI to predict the bioactive compounds in TFDM, potential protein targets and pathways. We also performed in vitro experiments to validate the potential underlying mechanism of TFDM on CIRI, as predicted by network pharmacology approach. To the best of our knowledge, this is the first time that a potential mechanism for the treatment of CIRI by TFDM has been studied through network pharmacology combined with *in vitro* experimental verification.

## 2. Materials and methods

### 2.1 Collection and screening of active components of total flavonoids

Using "Dracocephalum moldavica L." as the key word, the chemical components of dracocephalum moldavica were collected through literature retrieval, BATMAN-TCM database and ChemSrc database (https://www.chemsrc.com/). Potential active ingredients were screened in the SwissADME (http://www.swissadme.ch/) database based on pharmacokinetic parameters. Filter by [9, 10]: (1) defining gastrointestinal absorption (gastrointestinal absorption / GI absorption) in pharmacokinetics as "high" as a condition for chemical composition; (2) bio-availability score ≥ 0.5; (3) at least two of the pharmaceutical (druglikeness) rules as "yes"; (4) screening flavonoids active components.

### 2.2 Prediction and screening of targets

**2.2.1 Prediction of action target of total flavonoid extract of D.moldevica(TFDM).** The structural formula of the compound was imported into the Swiss Target Prediction (http://www.swisstargetprediction.ch/) database, and to obtain the potential action targets corresponding to TFDM.

**2.2.2 Collection of disease related targets.** The DrugBank (https://www.drugbank.ca/), OMIM (https://www.omim.org/), and GeneCards (https://www.genecards.org/) databases were searched with the keyword "Cerebral ischemia-reperfusion injury" to collect genes related to cerebral ischemia-reperfusion injury. In the Uniprot database, and to calibrate the disease-related gene target name to the Uniprot official gene name.

### 2.3 Construction and analysis protein-protein interaction (PPI) network

Obtaining PPI related information on an intersection target of an active component of the TFDM and CIRI by using a String (https://String-DB.org/) database to construct a PPI visual network; The PPI network was analyzed with Cytoscape 3.8.0 software, and after the topology

parameters were tested in the NetworkAnalyzer settings, the key targets of each component in TFDM against CIRI were identified and visualized in the network diagram [11].

## 2.4 Gene functional enrichment and analysis

Gene ontology(GO) and Kyoto Encyclopedia of Genes and Genomes (KEGG) enrichment analysis of the prevention and treatment of CIRI targets of TFDM were performed by online gene function enrichment analysis tool based on DAVID (https://david.ncifcrf.gov/) database. According to the data information, the corresponding rich factor was calculated, and the first 20 gene functions in ascending order of log(Pvalue) value were selected, with the name, enrichment factor, log(Pvalue) value and gene count as the data. The bubble charts of molecular function (MF), biological process (BP) and cellular components (CC) were obtained by visual mapping. The first 20 gene pathways in ascending order of log(Pvalue) value were selected to produce a bubble chart, and the 20 pathways were used as key pathways to create a signal pathway network diagram of TFDM anti-CIRI targets using Cytoscape 3.8.0 software.

## 2.5 Molecular docking

The active compound components in TFDM were molecular docked with protein kinase B (AKT1) and proto-oncogene tyrosine-protein kinase (SRC) respectively. The structures of related proteins were downloaded from the RCSB(https://www.rcsb.org/) database with PDB ID of 7NH5 and 7N9G. Solvent molecules and ligands were removed using Discovery Studio software, and hydrogenation and electron addition were performed using AutoDock Vina software. The compound structure was downloaded from PubChem(https://pubchem.ncbi.nlm.nih.gov) database, and hydrogenation, electron addition, and ROOT addition were performed using AutoDock Vina software. Molecular docking was performed after completion, in which the protein structure was set as a rigid macromolecule, and the first three groups with the lowest binding energy of each protein in the results were obtained. The best results were plotted using Pymol. Besides, the molecular docking structure and interaction relationship of the effective compound molecules with the AKT1 and SRC target proteins were examined by using Discovery Studiot software.

## 2.6 Pharmacological verification of network analysis

**2.6.1 Materials.** A total of 60 two-month-old male SPF SD rats, weighing (180±10) g, were purchased from Sxbex Biotechnology Company(Henan Province,China,LicenseNo.: SCXK (豫) 2020–0005);The total flavonoid extract of D.moldevica (TFDM) is made by the Xinjiang Uygur Autonomous Region Institute of Medicine (batch number: 20180412, 58.4%); SDS-PAGE gel preparation kit (P0012A, Beyotime Biotechnology Company, China), lactate dehydrogenase (LDH) release assay kit (A020-2-2, Nanjing Jiancheng Institute of Bioengineering, China), Nitric Oxide (NO) assay kit(A013-2-1, Nanjing Jiancheng Institute of Bioengineering, China), Malondialdehyde (MDA) assay kit (A003-4-1, Nanjing Jiancheng Institute of Bioengineering, China), IL-1β ELISA kit (E-EL-R0012c, Elabscience Biotechnology, China), IL-6 ELISA kit (E-EL-R0015c, Elabscience Biotechnology, China), TNF-α ELISA kit (E-EL-R2856c, Elabscience Biotechnology, China), BCA protein quantification kit (23225, Thermo Fisher Pierce, US), primary antibodies AKT(#4060, CST Company, US), p-AKT (#9272, CST Company, US), SRC(ab40660, Abcam, US), β-actin (ab8226, Abcam, US), GAPDH antibody(TA-08, Zhongshan Jinqiao Company, China), horseradish enzyme-labeled goat anti-mouse IgG (ZB-2305, Zhongshan Jinqiao Company, China), horseradish enzyme-labeled goat anti-rabbit IgG (ZB-2301, Zhongshan Jinqiao Company, China).

**2.6.2 Grouping, drug treatment and construction of model.** All animal experiments (Approval number: XJIMM-20211006) were reviewed and approved by the Ethics Committee of the Xinjiang Institute of Materia Medica. Animal welfare was monitored daily by animal care staff.The approval document (Ethics Statements) has been uploaded to other Item. Human participants and/or tissues were not involved in this study. After the rats are bought back, they are kept in the animal house for 7 days to adapt to the environment. After the rats adapt to the environment, the SD rats were randomly divided into a sham operation group, a model group, and TFDM 30, 60 and 120 mg groups, a total of 5 groups, 10 rats in each group. The rats of the three TFDM dose groups were intragastrically administered with TFDM 30, 60 and 120 mg·kg$^{-1}$ per day 7 d before surgery, respectively. The rats of the sham operation group and the model group were intragastrically administered with corresponding doses of normal saline, it lasted for 7 days, on the morning of the eighth day, the CIRI model was established two hours after the administration was completed, 45min after the operation caused cerebral ischemia, and 24 hours after reperfusion, it created a cerebral ischemia-reperfusion injury model, and the samples were taken 24 hours after operation.

Because of the demand of the experiment, serum is needed to complete the follow-up experiment, so this experiment adopts the method of acute bloodletting to sacrifice. The specific method is to cut the abdominal aorta of rats and take blood. The anesthesia method is: 2% pentobarbital sodium for anesthesia. In order to minimize the pain of rats, each anesthesia should wait for the rats to be deeply anesthetized before proceeding to the next step. If the rats are not deeply anesthetized, a certain amount of anesthetic will be supplemented to meet the requirements of deep anesthesia.

Establishment of cerebral ischemia-reperfusion injury model using middle cerebral artery suture method [12]; Rats were anesthetized first; An incision was made in the middle of the neck to separate the left common carotid artery, internal carotid artery and external carotid artery. Ligate common carotid artery and external carotid artery; A small incision was cut below the bifurcation of the common carotid artery. The 0.3 mm nylon thread was inserted into the internal carotid artery until it met with slight resistance, and then it was stopped. The nylon thread was fixed, and the thrum was gently pulled back for reperfusion after 45min of embolization. In the sham operation group, monofilament nylon suture was not used to block the middle artery, and other treatments were the same as those in the model.

**2.6.3 Oxidative stress response detection.** Blood was taken from the abdominal aorta of rats with a blood collection needle. The blood collection tube was put into a high-speed centrifuge for centrifugation for 10min, and the supernatant was taken for later use. The contents of LDH, MDA and NO were measured according to the kit instructions.

**2.6.4 Inflammatory response detection.** IL-6, IL-1β and TNF-α contents were determined according to the kit instructions by ELISA.

**2.6.5 Detection of protein expression in brain tissue by western blotting.** Extracting total protein from brain tissue, and detecting protein concentration by using a BCA kit;The protein to be tested was separated by polyacrylamide gel electrophoresis, transferred to PVDF membrane, and then blocked in the blocking solution for 2 h; Next, a corresponding primary antibody [AKT(1:1000), p-AKT(1:1000), SRC(1:1000), β-actin(1:1000)] is added, overnight at 4°C, then a horseradish peroxidase (HRP)-labeled secondary antibody was added. Incubated at room temperature for 2 h and finally imaged in a dark room with chemiluminescent reagent.

**2.6.6 Statistical analysis.** Statistical analysis was performed with GraphPad Prism 8.0 software. All data were expressed as the mean ± standard deviation (SD). The differences among multiple groups were evaluated using the one-way analysis of variance (ANOVA). The difference between the means was considered statistically significant at P < 0.05.

**Table 1. Active compounds screened from TFDM.**

| NO. | Compound Name | PubChem CID |
|---|---|---|
| HT01 | 8-hydroxy-salvigenin | 3083783 |
| HT02 | scrophulein | 188323 |
| HT03 | acacetin | 5280442 |
| HT04 | apigenin | 5280443 |
| HT05 | luteolin | 5280445 |
| HT06 | chrysoeriol | 5280666 |
| HT07 | diosmetin | 5281612 |
| HT08 | gardenin A | 261859 |
| HT09 | gardenin B | 96539 |
| HT10 | isorhamnetin | 5281654 |
| HT11 | kaempferol | 5280863 |
| HT12 | quercetin | 5280343 |
| HT13 | salvigenin | 161271 |
| HT14 | syringaresinol | 100067 |
| HT15 | 3-hydroxyflavone | 11349 |
| HT16 | moldavoside, acacetin 7-O-glucoside | 5321954 |
| HT17 | apigenin-7-O-β-D-galactoside | 44257799 |
| HT18 | acacetin7-O-β-D-glucopyranoside | 44257884 |
| HT19 | acacetin-7-O (- 6"-acetyl) -glucopyranoside | 52929806 |
| HT20 | acacetin-7-O-β-D-glucuronide | 44257886 |
| HT21 | apigenin-7-O-β-D-glucoside | 5280704 |

## 3. Results

### 3.1 Preparation of active compounds of TFDM

The active components of TFDM were identified using the SwissADME ™ analysis platform and literature collection. To ensure the accuracy and completeness of the data, a small number of bioactive molecules were added as candidate active molecules according to the literature reports, although they did not meet the screening criteria. Finally, 21 compounds were selected as candidate active molecules, and the results are shown in Table 1.

### 3.2 Target prediction and screening

Target prediction was performed on 21 active compounds in TFDM, and the duplicate values were deleted after combination, so that a total of 158 human source targets were obtained. For the targets related to cerebral ischemia-reperfusion injury, the data were collected by Drug-Bank, OMIM and GeneCards databases, and the disease targets collected by the three databases were combined for deduplication, and a total of 797 disease targets related to cerebral ischemia-reperfusion injury were obtained. Intersecting the drug targets of the TFDM active compounds with the disease targets related to CIRI and preparing a Venn diagram, obtaining 53 potential action targets of TFDM for preventing and treating CIRI, and importing the targets into a STRING database to obtain interaction relationship data between the targets and visualize a PPI network, as shown in Fig 1.

### 3.3 Analysis of main targets of PPI network

The interaction relationship data between 38 main targets and 51 groups of targets with combined_score≥0.9 were selected, and the subnet cluster obtained by MCODE algorithm in Cytoscape software was shown in Fig 2.

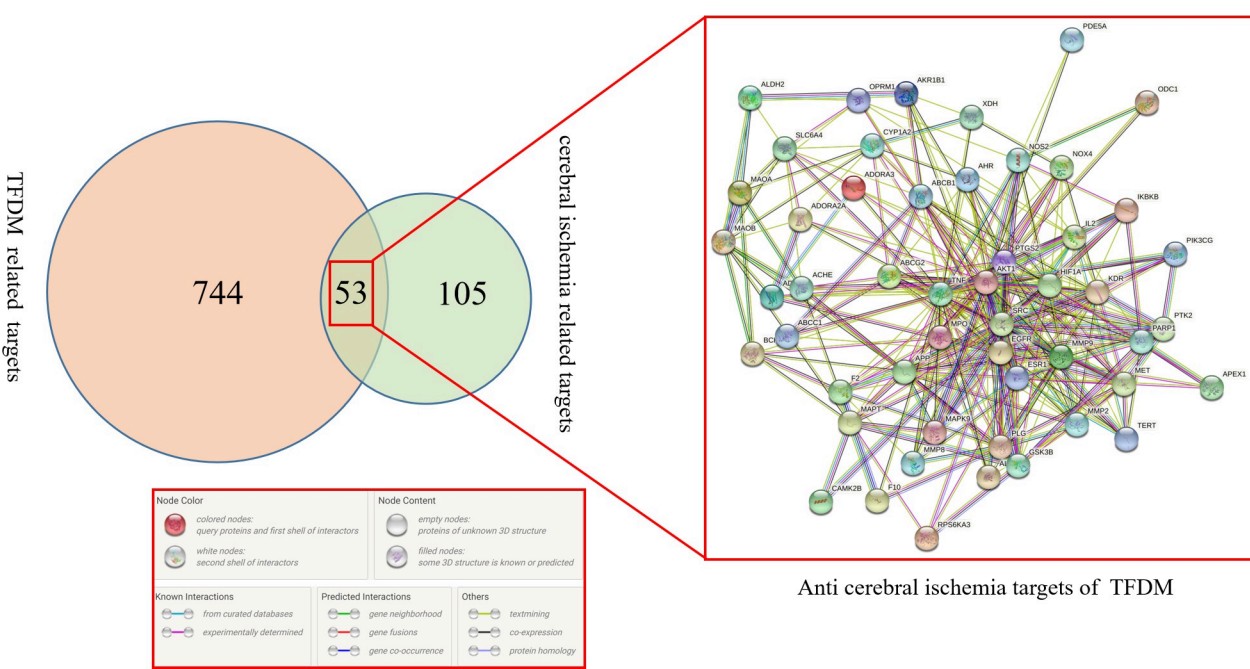

Anti cerebral ischemia targets of TFDM

**Fig 1. Venn diagram shows all the common merged targets for TFDM and CIRI.**

Among them, predicted targets IL2, AKT1, PTGS2, HIF1A, MMP9, MET, EGFR, ESR1, TNF, SRC, APP, PLG, KDR and MP2 were clustered as one class, targets MPO, MMP8 and F2 were categorized into one class, and targets CYP1A2, MAOB and SLC6A4 were categorized into another class.

Screening of key targets based on topological data, degree values and eigenvector centrality value resulted in the screening of six CIRI-associated TFDM targets, including SRC, AKT1, ESR1, EGFR, HIF1A, and NOS2. The importance of the visualized target was represented by the degree value and eigenvector centrality value, as shown in Fig 3.

## 3.4 Results of GO enrichment analysis and KEGG pathway analysis

Using DAVID database for GO enrichment analysis of potential targets, a total of 152 GO entries with P<0.05 were obtained, including 76 entries for biological process (BP), 26 entries

PPI Network of Cluster of Anti cerebral ischemia of TFDM

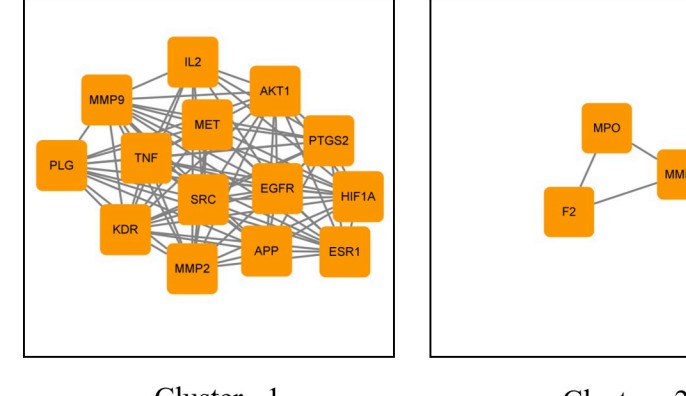

Cluster 1

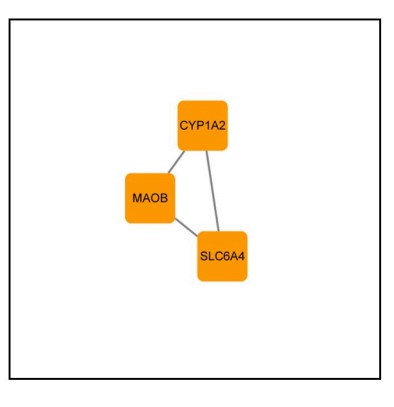

Cluster 2            Cluster 3

**Fig 2. Subnetwork clusters of identified targets for TFDM against CIRI obtained using MCODE algorithm.**

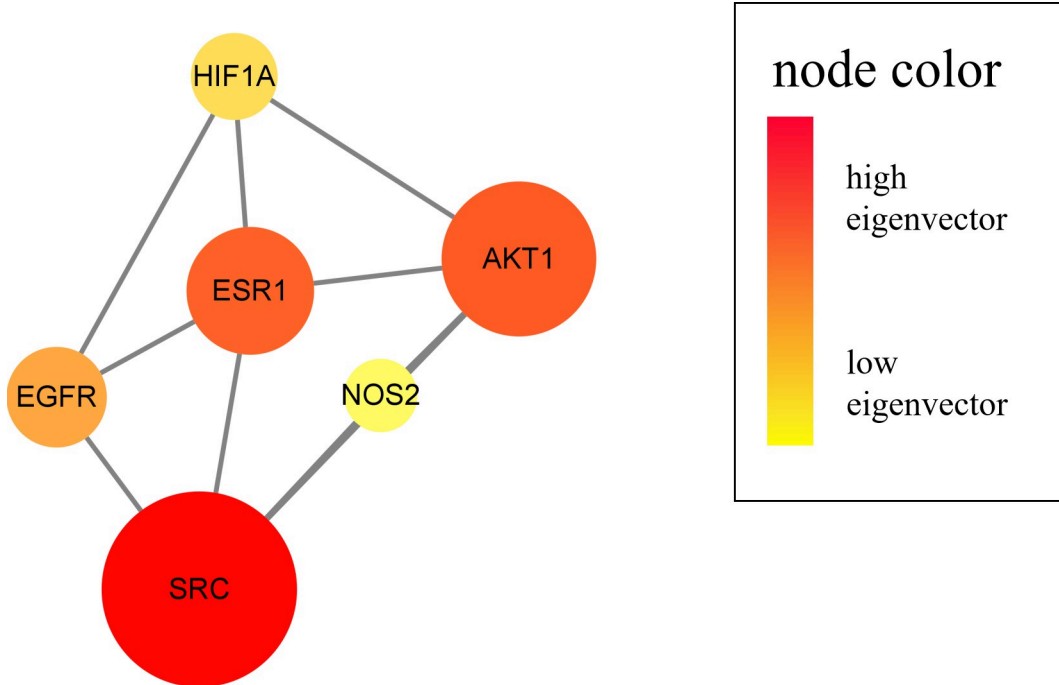

**Fig 3. CIRI-related pivotal targets of TFDM.** Six pivotal targets were screened and identified from merged targets, namely SRC, AKT1, ESR1, EGFR, HIF1A, and NOS2.

for cell composition (CC), and 50 entries for molecularfunction (MF). The pathways in the top 20 of each class are shown in Fig 4. Through KEGG pathway analysis, a total of 106 pathways (P<0.05) were identified, including PI3K-AKT signaling, pathwayTNF signaling pathway et al. [13] reported related to CIRI, as well as ErbB signaling pathway, Relaxin signaling pathway, EGFR tyrosine kinase inhibitor resistance and other pathways related to CIRI protection. Where the top 22 pathways are shown in Fig 4.

## 3.5 Construction and analysis of compound-target-pathway network

The first 22 signaling pathways obtained from KEGG analysis, as well as the enriched key targets and corresponding compound data were imported into Cytoscape software to construct a weighted network of "compounds-targets-pathways". As shown in Fig 5, a green rectangle indicated 22 key pathways, an orange diamond indicated 24 key targets, a yellow circle indicated 21 active compounds corresponding to the targets, the size of the points indicated degree value, and the color depth of the points indicated eigenvector centrality value. The deeper the color, the more important the node became.

## 3.6 Molecular docking results of the effective compound components in TFDM with the two proteins

Molecular docking resulted in the binding energy results of 21 active components in TFDM with AKT1 and SRC, as shown in Table 2.

The docking results of the top 3 compounds that had the best docking binding per protein were visualized.The binding energy of AKT1 protein (7NH5) to the original binding ligand UC8502 was -12.8 kcal/mol, and its binding energies to the compounds HT-19, HT-18 and HT-17 were -11.0 kcal/mol, -10.9 kcal/mol and -10.8 kcal/mol, respectively. In docking with the AKT1 protein, HT19 was found to form hydrogen bonds with amino acid residues of

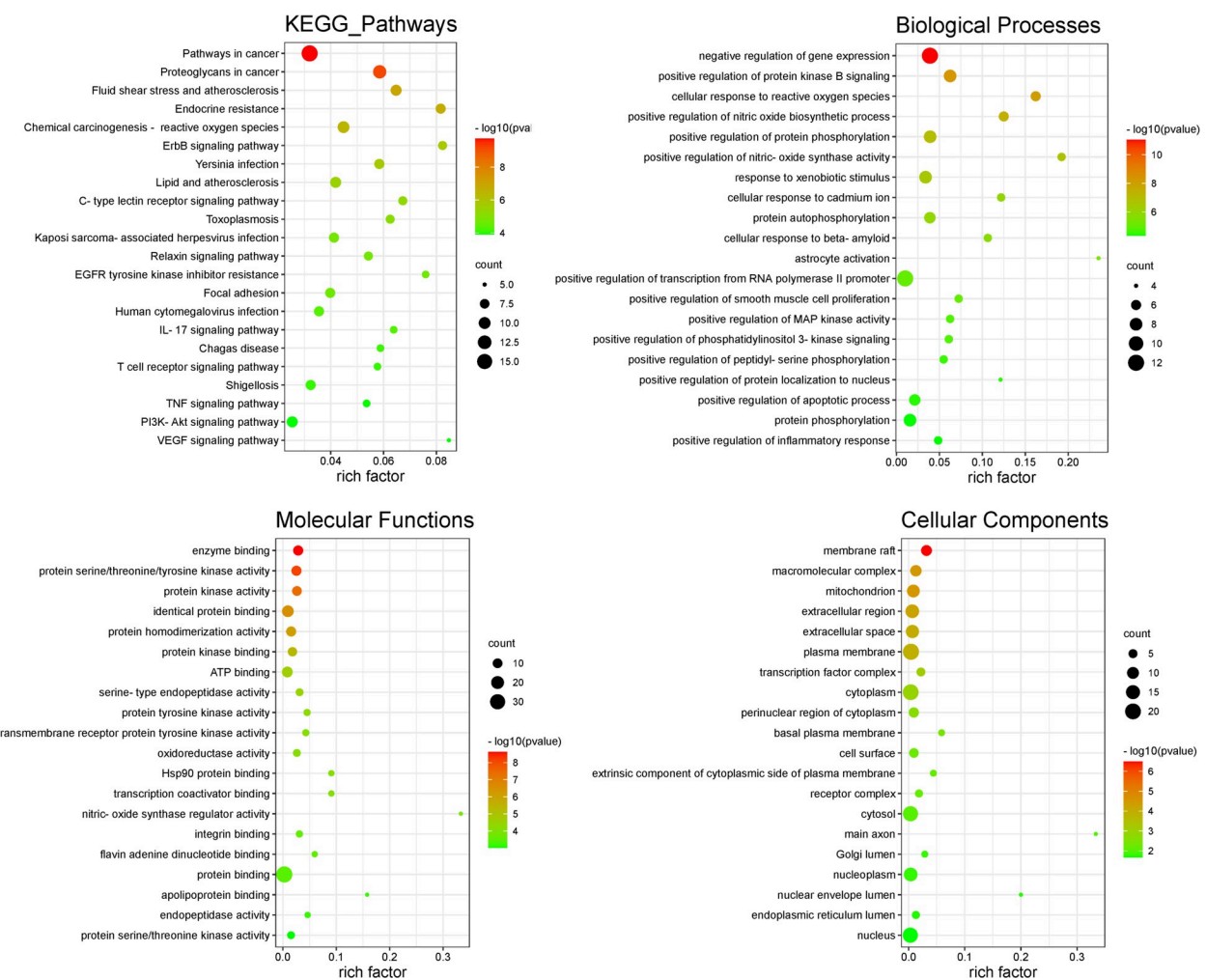

**Fig 4. KEGG enrichment analysis and GO enrichment analysis.**

LYS-A20 (2.20Å), ASN-A54 (2.26Å), THR-A82 (2.14Å), LYS-A179 (2.51Å), GLY-A294 (2.96Å), and CYS-A296 (3.55Å) (Fig 6A); HT18 was found to form hydrogen bonds with amino acid residues of GLU-A17 (3.37Å), LYS-A20 (2.22Å), THR-A82 (2.42Å), LYS-A179 (2.44Å), and GLY-294 (2.27Å) (Fig 6B); HT17 was found to form hydrogen bonds with the amino acid residues of ASN-A54 (2.92Å), LYS-268 (3.65Å), and TYR-272 (2.66Å) (Fig 6C).

The binding energy of SRC protein (7N9G) to the original binding ligand STI601 is -9.6 kcal/mol, and its binding energy to the compounds HT-19, HT-20 and HT-18 is -9.0 kcal/mol, -8.9 kcal/mol and -8.8 kcal/mol, respectively. In docking with SRC protein, HT19 was found to form hydrogen bonds with amino acid residues of ALA-A356 (2.70Å), TYR-A454 (2.49Å) and GLU-A481 (2.23Å) (Fig 7A); HT20 was found to form hydrogen bonds with amino acid residues of ALA-A452 (2.49), TYR-A454 (2.21Å) and GLY-A482 (2.35Å) (Fig 7B); HT18 was found to form hydrogen bonds with the amino acid residues of GLU-A481 (2.13Å) (Fig 7C).

### 3.7 Experimental validation

**3.7.1 Results of oxidative stress.** Compared with the sham operation group, the MDA and LDH contents in the model group were significantly increased, and the NO content was

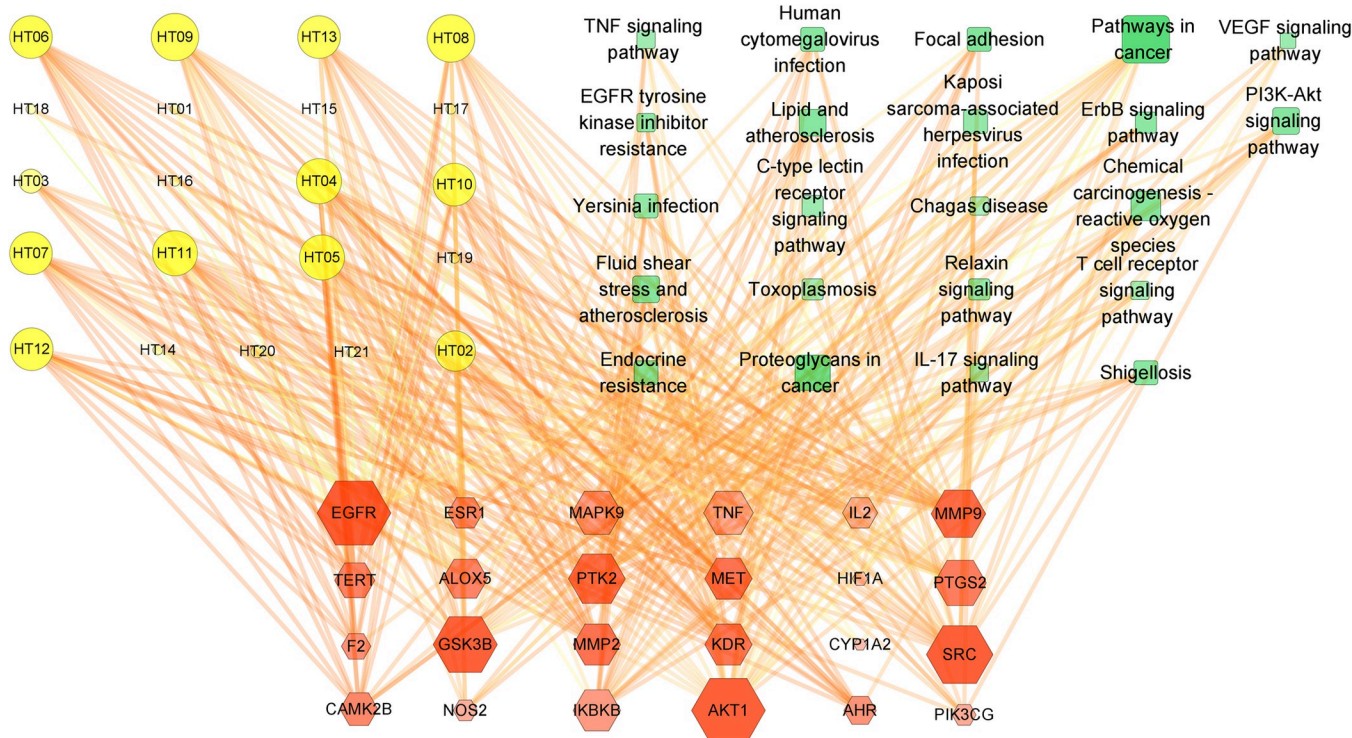

**Fig 5. Network of compound-target-pathway.**

**Table 2. Docking results of effective compounds in TFDM.**

| NO. | AKT1 (kcal/mol) | Sort order | SRC (kcal/mol) | Sort order |
|---|---|---|---|---|
| HT01 | -9.9 | 9 | -7.3 | 19 |
| HT02 | -9.4 | 18 | -7.7 | 13 |
| HT03 | -9.7 | 10 | -7.8 | 12 |
| HT04 | -10.2 | 8 | -8.0 | 8 |
| HT05 | -9.5 | 14 | -7.9 | 10 |
| HT06 | -9.5 | 15 | -8.0 | 9 |
| HT07 | -9.1 | 20 | -7.7 | 14 |
| HT08 | -8.7 | 21 | -6.5 | 21 |
| HT09 | -9.6 | 12 | -7.1 | 20 |
| HT10 | -9.7 | 11 | -7.6 | 15 |
| HT11 | -9.5 | 13 | -7.9 | 11 |
| HT12 | -9.4 | 17 | -7.5 | 16 |
| HT13 | -9.5 | 16 | -7.4 | 17 |
| HT14 | -10.4 | 6 | -7.3 | 18 |
| HT15 | -10.6 | 5 | -8.6 | 5 |
| HT16 | -10.8 | 4 | -8.2 | 6 |
| HT17 | -10.8 | 3 | -8.7 | 4 |
| HT18 | -10.9 | 2 | -8.8 | 3 |
| HT19 | -11.0 | 1 | -9.0 | 1 |
| HT20 | -10.4 | 7 | -8.9 | 2 |
| HT21 | -9.3 | 19 | -8.1 | 7 |
| Ligand | -12.8 | / | -9.6 | / |

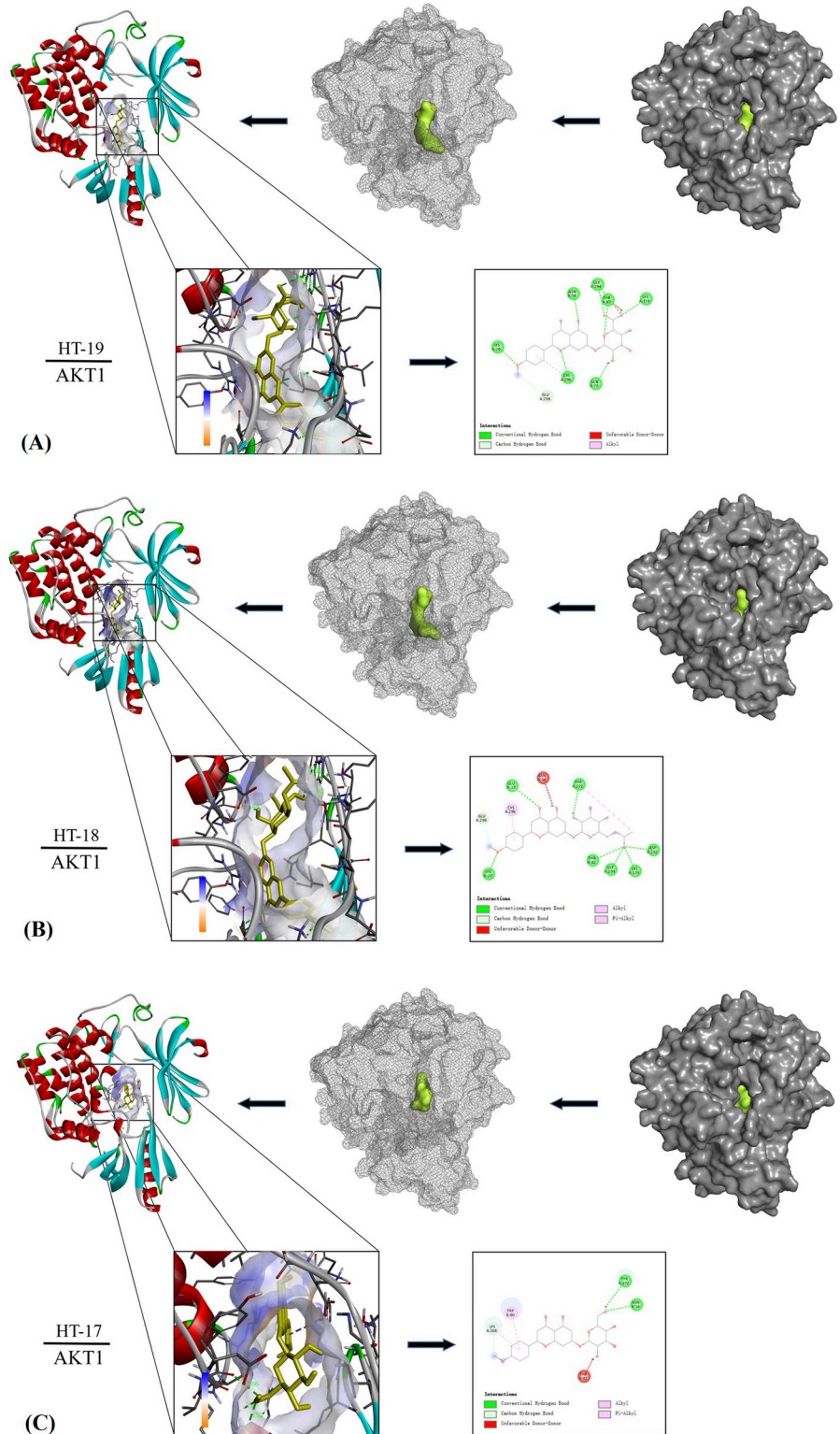

**Fig 6. Compounds HT19, HT18 and HT17 respectively formed different numbers of hydrogen bonds with protein kinase B(AKT1).**

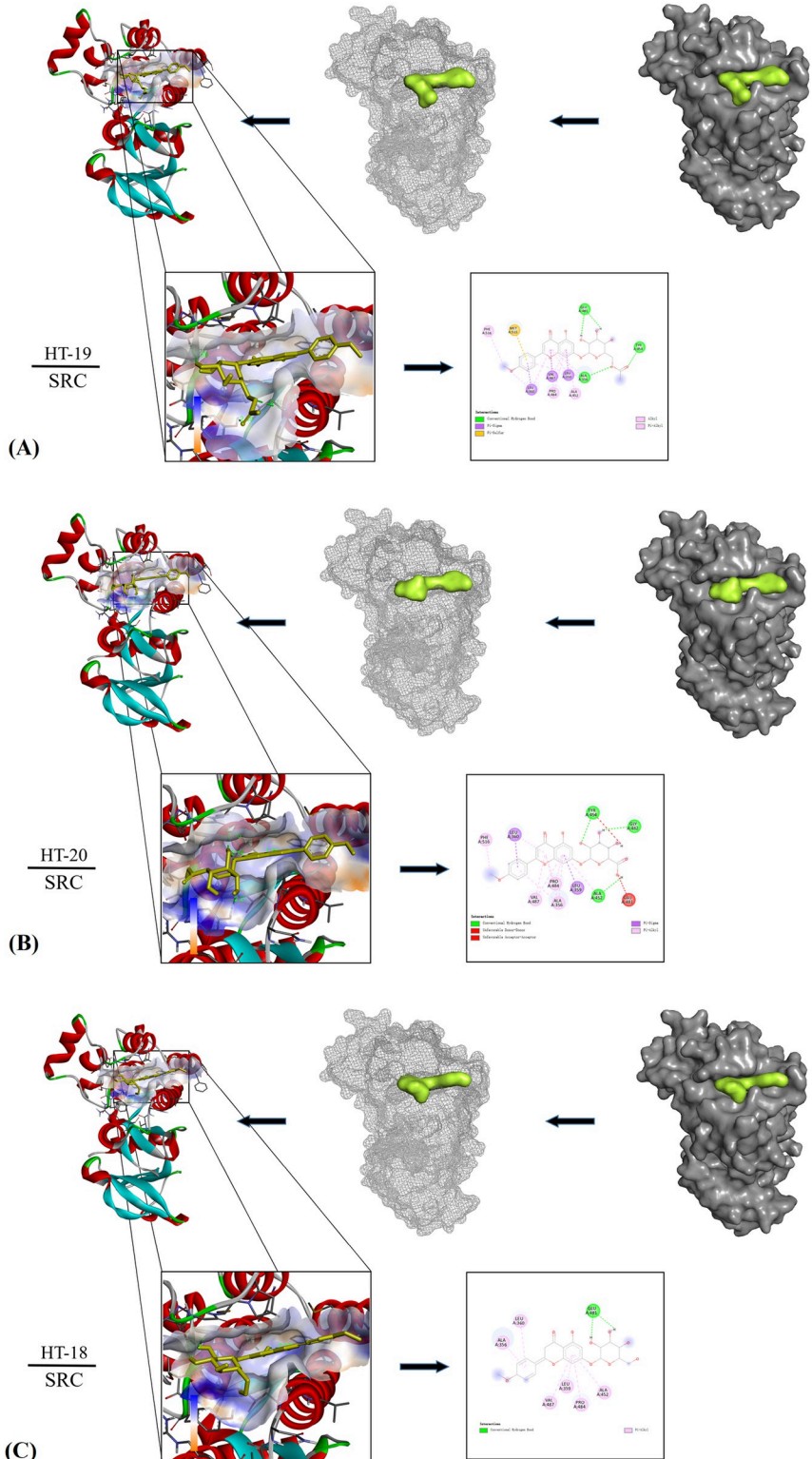

**Fig 7. The compounds HT19, HT20 and HT18 respectively formed different numbers of hydrogen bonds with tyrosine kinase (SRC).**

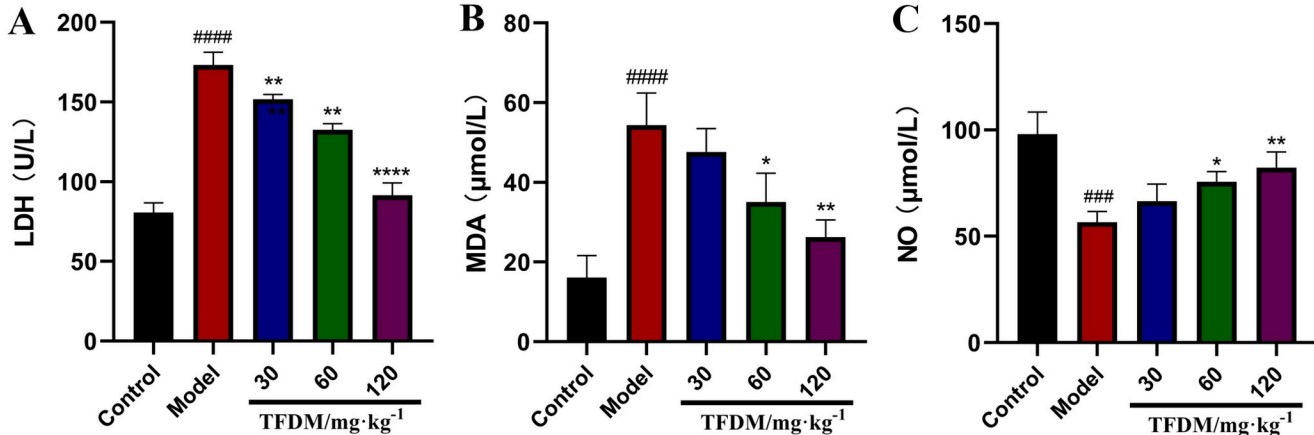

**Fig 8.** Effects of TFDM on the Changes of LDH (A), MDA (B) and NO (C) in CIRI. (A, n = 3, ####P <0.0001 versus the control group, **P<0.01 versus the model group, ****P<0.0001 versus the model group. B, n = 3, ####P <0.0001 versus the control group, *P<0.05 versus the model group, **P<0.01 versus the model group. C, n = 3, ###P <0.001 versus the control group, *P<0.05 versus the model group, **P<0.01 versus the model group).

significantly decreased (all P < 0.05). Compared with the model group, the MDA and LDH contents were significantly decreased and NO content was significantly increased in the TFDM 30, 60 and 120 mg groups. With the increase of the drug dose, the effect was enhanced, as shown in Fig 8.

**3.7.2 Results of inflammatory response.** Compared with the sham operation group, the IL-1β, IL-6 and TNF-α contents in the model group were significantly increased (P < 0.05). Compared with the model group, the contents of IL-1β, IL-6 and TNF-α in the TFDM 30, 60 and 120 mg groups were significantly reduced, and the effect was enhanced with the increase of the drug dose, as shown in Fig 9.

**3.7.3 Effects of TFDM on the expression of AKT, SRC and other related proteins.** Compared with the control group, the expression levels of p-AKT/AKT and SRC/β-acttin in the model group were significantly down-regulated (P < 0.05). Compared with the model group, the p-AKT/AKT and SRC/β-acttin expressions in the 30, 60, and 120 mg groups were

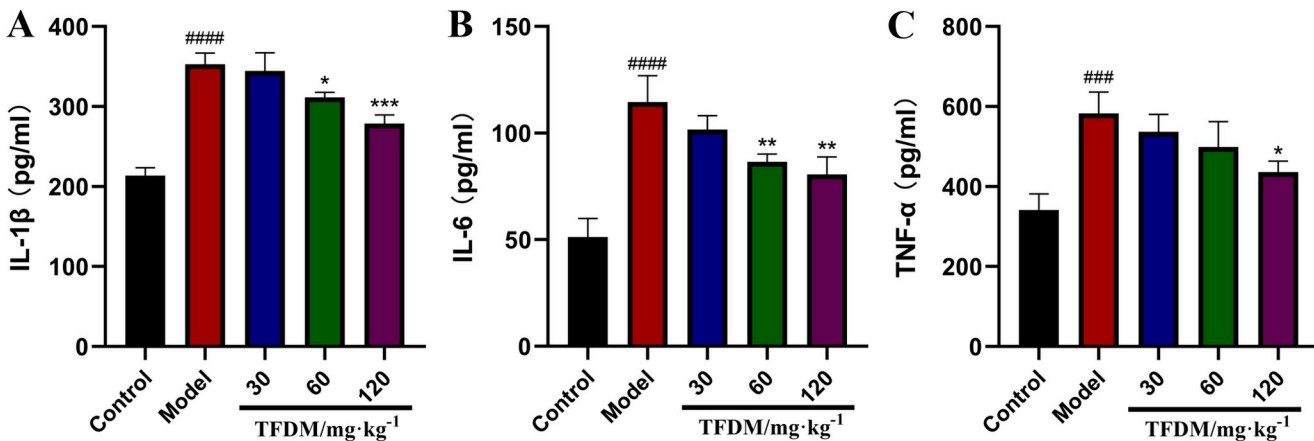

**Fig 9.** Effects of TFDM on the Changes of IL-1β (A), IL-6 (B) and TNF-α (C) in CIRI. (A, n = 3, ####P <0.0001 versus the control group, *P<0.05 versus the model group, ***P<0.001 versus the model group. B, n = 3, ####P <0.0001 versus the control group, **P<0.01 versus the model group. C, n = 3, ###P <0.001 versus the control group, *P<0.05 versus the model group).

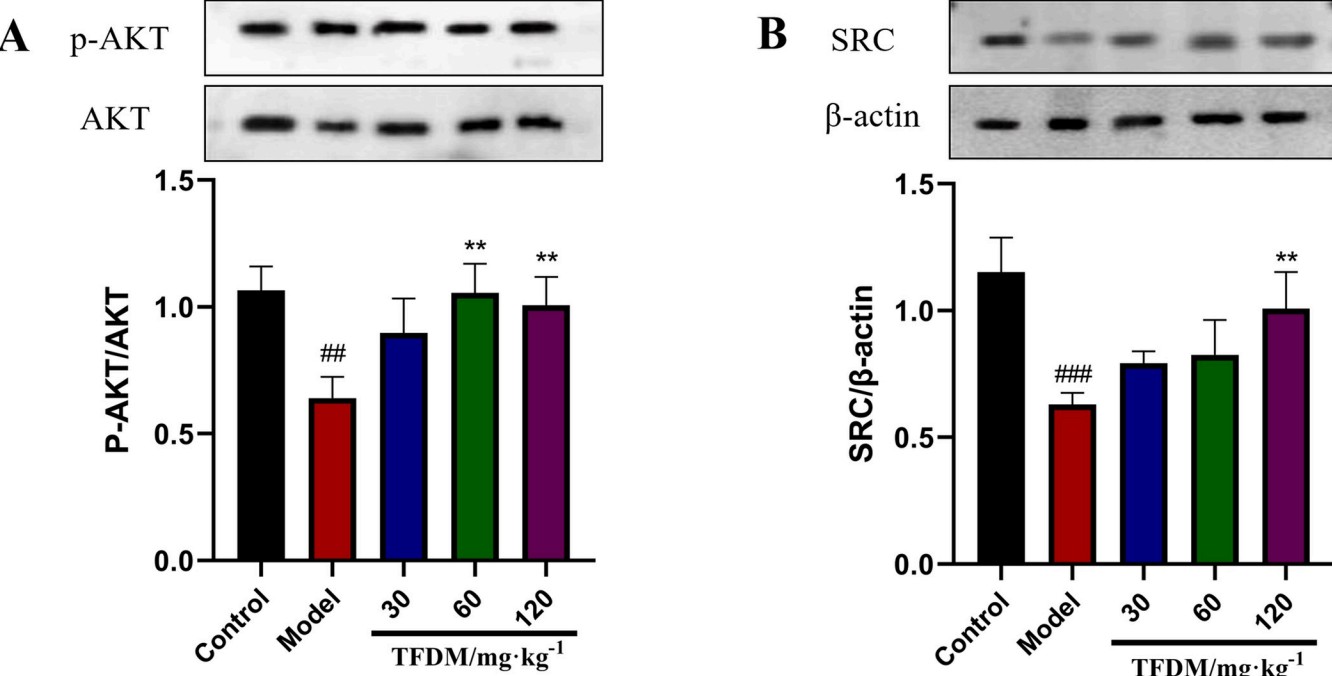

**Fig 10. Expression of pathway proteins was detected by western blot analysis.** (A, n = 3, ##P <0.01 versus the control group, **P<0.01 versus the model group. B, n = 3, ###P <0.001 versus the control group, **P<0.01 versus the model group).

significantly up-regulated (S1 Raw images), and the effect was enhanced with the increase of the dosage, as shown in Fig 10.

## 4. Discussion

Cerebral ischemia-reperfusion injury(CIRI) is a common and frequently-occurring disease in the clinic. The mortality and disability rate are extremely high. It seriously threatens human health and life, and the burden on society and families is immeasurable. Therefore, scientists and clinicians need to make great efforts to find potential pathogenic mechanisms, so as to take appropriate preventive measures and successful treatment methods. CIRI involves a variety of mechanisms [14], including free radical action, excitatory amino acids, cellular $Ca^{2+}$ overload. To understand the mechanism of CIRI and deepen the understanding of the process of cerebral ischemic injury is conducive to the research and selection of medication. so as to achieve the purpose of treatment. Natural medicines have a long history of treatment for cerebral ischemia and have accumulated rich theoretical knowledge. Screening active ingredients from natural medicines is also the current trend of research and development. The whole world is promoting the separation of chemical components of natural medicines and their application in the treatment of brain diseases. Our research group has continuously conducted a series of studies on this issue, and have made great progress. This study found that TFDM has unique advantages and potential in the treatment of CIRI because of its diverse chemical components and structures, and can intervene the pathological process of CIRI through multiple targets and channels as a whole. In this paper, TFDM pharmacodynamics and its mechanism of action are taken as the breakthrough point, in order to provide theoretical basis for drug research and development of CIRI and scientific basis for in-depth research on prevention and treatment of CIRI in traditional Chinese medicine.

In this study, 21 active compounds that can be absorbed by blood were found in TFDM by network pharmacology. These 21 compounds have 53 potential targets for the treatment of cerebral ischemia-reperfusion injury. Using bioinformatics analysis on the interaction relationship data between 38 main targets of combined_score≥0.9 and 51 groups of targets, the obtained results revealed the key targets, biological functions and molecular pathways of TFDM participating in the treatment of CIRI. We finally focused on two key targets for the treatment of CIRI, AKT1 and SRC; Among the first 22 signaling pathways enriched in this experiment, SRC was involved in the regulation of 15 signaling pathways, and AKT1 was involved in the regulation of almost all signaling pathways. Tyrosine protein kinase SRC is activated after binding to different types of cellular receptors (including immunoreaction receptors, integrins and other adhesion receptors, receptor protein tyrosine kinases, G protein-coupled receptors, and cytokine receptors). The activated SRC promotes cell angiogenesis by increasing the expression of angiogenic factors such as VEGF and IL-18. Akt has three subtypes, AKT1, AKT2 and AKT3. Among them, AKT1 is a subtype that is highly expressed in the brain tissue, accounting for 70%–80% of the total AKT [15], and plays a regulatory role in anti-apoptosis and promotion of cell differentiation, proliferation, migration and cell metabolism. Under general conditions, Akt of stationary cells is mostly located in the cytoplasm and does not work. When the brain tissue is stimulated by ischemia and hypoxia, p-Akt1, in addition to inhibiting apoptosis and reducing ischemia-reperfusion injury as the key factor against apoptosis in PI3K/Akt signaling pathway, can regulate cell cycle, promote cell survival and angiogenesis and other various cell activities and biological effects by phosphorylating its downstream protein factors [16]. Recent studies have shown that after phosphorylation of Akt, nitric oxide synthase can be activated to promote neoangiogenesis and make the new blood vessels gradually extend from the edge of the infarction to the ischemic central area, in order to restore blood supply to the ischemic area.

When cerebral ischemia-reperfusion occurs, ischemia triggers the production of a large number of reactive oxygen species [17], and the inflammatory response is accompanied by the persistence of CIRI. The experimental results have shown that TFDM can significantly alleviate the adverse effects of CIRI-induced cellular inflammation and oxidative stress pathological process. PI3K-Akt autophagy pathway is closely related to the occurrence of ischemia-reperfusion injury. Some researchers found that hesperidin played a protective role in rats with myocardial ischemia-reperfusion injury by inhibiting autophagy through PI3K-Akt pathway [18]. LI et al. [19] reported that tanshinone IIA improved myocardial ischemia-reperfusion injury through PI3K-Akt signaling pathway. Therefore, regulating PI3K-Akt signaling pathway can improve ischemia-reperfusion injury. In this study, we found that TFDM could significantly up-regulate the expression of p-Akt and SRC in CIRI rats, and the effect showed an increasing trend with the increase of the drug dose.

Molecular docking analysis showed that most of the active compounds in TFDM had good binding affinity with the targets (AKT1 and SRC). Although the binding force of these compounds was not the strongest compared with the existing ligands of the target, it was very close, especially SRC. These findings strongly indicated the role of TFDM in the treatment of CIRI.

In summary, our bioinformatics research and experimental validation have demonstrated that TFDM contributes to the inhibition of neuroinflammation and neuropathy/necrosis, and is a promising therapeutic strategy for CIRI. However, one of the shortcomings of traditional Chinese medicine in treating diseases is its slow onset, and the rapid onset of chemical drugs is its advantage over traditional Chinese medicine. Therefore, the rational combination of traditional Chinese medicine and chemical drugs can be considered at present, which provides a new direction for anti-CIRI research.

In the future, we will conduct further research on the active components in TFDM that have strong binding affinity with key targets.We believe that with the deepening of the research on the pathogenesis of CIRI, the research on the mechanism of action, treatment time window and pharmacokinetics of TFDM and its compound anti-CIRI is becoming increasingly clear, and the current situation of CIRI treatment difficulties will certainly make a breakthrough.

## Supporting information

**S1 Raw images. It provide the original underlying images for western blot analysis of Fig 10.**
(TIF)

## Acknowledgments

We sincerely thank the team of aid experts in Xinjiang for their help and technical support for this study.

## Author Contributions

**Conceptualization:** Xu Hu, Rui-fang Zheng, Jian-guo Xing.

**Data curation:** Xu Hu, Yideresi Mola.

**Formal analysis:** Xu Hu, Yideresi Mola, Yue Wang.

**Funding acquisition:** Xu Hu, Rui-fang Zheng, Jian-guo Xing.

**Investigation:** Yideresi Mola, Wen-ling Su, Yue Wang.

**Methodology:** Xu Hu, Rui-fang Zheng.

**Project administration:** Jian-guo Xing.

**Resources:** Wen-ling Su, Yue Wang.

**Software:** Yideresi Mola.

**Supervision:** Jian-guo Xing.

**Validation:** Yideresi Mola, Wen-ling Su, Yue Wang.

**Visualization:** Wen-ling Su.

**Writing – original draft:** Xu Hu.

**Writing – review & editing:** Rui-fang Zheng.

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
