## [Decision Letter · Decision Letter 0]

24 Apr 2023

PONE-D-22-35325A network pharmacology
approach to decipher the total flavonoid extract of Dracocephalum Moldavica L. in
the treatment of cerebral ischemia-
reperfusion injuryPLOS ONE

Dear Dr. Hu,

Thank you for submitting your manuscript to PLOS ONE. After careful consideration, we
feel that it has merit but does not fully meet PLOS ONE’s publication criteria as it
currently stands. Therefore, we invite you to submit a revised version of the
manuscript that addresses the points raised during the review process.

Please submit your revised manuscript by Jun 03 2023 11:59PM. If you will need more
time than this to complete your revisions, please reply to this message or contact
the journal office at plosone@plos.org. When
you're ready to submit your revision, log on to https://www.editorialmanager.com/pone/ and select the 'Submissions
Needing Revision' folder to locate your manuscript file.

Please include the following items when submitting your revised
manuscript:A rebuttal letter that responds to each point raised by the academic
editor and reviewer(s). You should upload this letter as a separate file
labeled 'Response to Reviewers'.A marked-up copy of your manuscript that highlights changes made to the
original version. You should upload this as a separate file labeled
'Revised Manuscript with Track Changes'.An unmarked version of your revised paper without tracked changes. You
should upload this as a separate file labeled 'Manuscript'.If you would like to make changes to your financial disclosure,
please include your updated statement in your cover letter. Guidelines for
resubmitting your figure files are available below the reviewer comments at the end
of this letter.

We look forward to receiving your revised manuscript.

Kind regards,

Yong Sze Ong

Academic Editor

PLOS ONE

Journal Requirements:

3. To comply with PLOS ONE submissions requirements, in your Methods section, please
provide additional information regarding the experiments involving animals and
ensure you have included details on (1) methods of sacrifice, (2) methods of
anesthesia and/or analgesia, and (3) efforts to alleviate suffering.

4. Please provide additional details regarding participant consent. In the ethics
statement in the Methods and online submission information, please ensure that you
have specified (1) whether consent was informed and (2) what type you obtained (for
instance, written or verbal, and if verbal, how it was documented and witnessed). If
your study included minors, state whether you obtained consent from parents or
guardians. If the need for consent was waived by the ethics committee, please
include this information.

Reviewers' comments:

Reviewer's Responses to Questions

**Comments to the Author**

1. Is the manuscript technically sound, and do the data support the conclusions?

Reviewer #1: Yes

Reviewer #2: Yes

Reviewer #3: Yes

2. Has the statistical analysis been performed
appropriately and rigorously? 

Reviewer #1: Yes

Reviewer #2: Yes

Reviewer #3: No

3. Have the authors made all data underlying the
findings in their manuscript fully available?

Reviewer #1: Yes

Reviewer #2: Yes

Reviewer #3: Yes

4. Is the manuscript presented in an intelligible
fashion and written in standard English?

Reviewer #1: Yes

Reviewer #2: Yes

Reviewer #3: No

5. Review Comments to the Author

Reviewer #1: The paper presented investigates the therapeutic use of Dracocephalum
moldavica L. (TFDM) in the cerebral ischemic reperfusion (CIRI) rat model. an
invasive ischemic rat model was established to validate the effects of the treatment
with TFDM. VEGF/SRC and PI3K/AKT pathways were identified as possible mechanisms
associated with CIRI. TFDM active ingredients and CIRI targets were identified by
thorough screening of multiple databases like Drugbank, Genecards and OMIM. Overall,
the paper clearly showed potential mechanisms that could be modulated by TFDM in
CIRI conditions.

Nevertheless, some points need more clarification like:

Why only male rats were chosen for the study?Lot or CAT numbers for the kits and antibodies used throughout the study
were not mentioned, although it’s crucial for such a study.After blood collection and centrifugation, how was the supernatant
stored? Also, how would the storage conditions affect LDH, MDA and NO
levels?What kind of effect is being validated by administering TFDM before the
surgery not just after the surgery?What was the success rate of establishing the CIRI rat model? As it’s
known such invasive procedure has a mortality rate. Were all the rats
assigned from the beginning alive throughout the rest of the study?As stated in the study, TFDM shows promise as a therapeutic strategy for
CIRI. Would that indicate the use of TFDM as a prophylactic in general
prior to the condition or after the ischemic injury?

Reviewer #2: Your research on the flavonoid extract of D. moldavica plant is
interesting and potentially valuable to the field.After carefully reading your work,
I would like to provide some constructive feedback.

1.Animal ethical Committee number should be mentioned in the manuscript.

2.The grouping of animals and drug treatment pattern is not clear.

3.I note that there have been similar research works published in the past. To
address this issue, I recommend that you provide a clear explanation of how your
current research builds upon and/or differs from the existing literature.

4.some of the references are not as per the journal guidelines

Reviewer #3: Authors evaluated the efficacy of total flavonoid extract of
Dracocephalum Moldavica L. in the treatment of cerebral ischemia- reperfusion injury
(CIRI) by using computational tools. This study describes the effectiveness of total
flavonoid (acacetin, apigenin etc.) to cure SIRI by significantly altering the
multi- channel signaling pathways. The study is quite interesting and useful as CIRI
endangers the human health with high mortality ratio. However, I suggest following
minor changes to improve the quality of the manuscript for readers and research
community.

Comments:

1. In Abstract section line 18, authors should replace “Disease” with “injury” as
cerebral ischemia- reperfusion injury (CIRI) is usually caused by caused by ischemic
stroke seriously or restriction of blood supply. Line 28, 90, in vitro should be
written in Italic. Abstract is lack of key findings of the study.

2. The ethical approval number should be added in methodology section as authors
mentioned the study has received approval from the Ethics Committee of the Xinjiang
Institute of Materia Medica.

3. Safety profile of total flavonoid extract of Dracocephalum Moldavica L. should be
addressed.

4. Please add software specifications used in statistical analysis (Graph prism).
More robust & appropriate statistical analysis is also required.

5. Limitation of the study should be discussed in limitation section of the
manuscript.

6. Authors should describe the Future perspective and clinical significance of the
study.

7. Authors should add abbreviations list in the manuscript.

6. PLOS authors have the option to publish the peer
review history of their article (what does this mean?). If published, this will
include your full peer review and any attached files.

If you choose “no”, your identity will remain anonymous but your review may still be
made public.

**Do you want your identity to be public for this peer review?** For
information about this choice, including consent withdrawal, please see our
Privacy Policy.

Reviewer #1: No

Reviewer #2: **Yes: **Dr. Fazil Ahmad

Reviewer #3: **Yes: **Mehmood Ahmad

---

## [Author Response · Author response to Decision Letter 0]

7 Jun 2023

Dear editors and peer reviewers,

First of all, it's a great honor to get your valuable suggestions. I attach great
importance to your review conclusion and will revise it according to your
suggestions to make the article more complete and rigorous. The following are my
answers to your Questions, hoping to get your approval.

Responses to general guidelines

1. Edited manuscript to meet PLOS ONE's style requirements, including the file
name.

2. Added (1) methods of sacrifice, (2) methods of anesthesia and analgesia.

3. We moved the ethics statement to the Methods section of the manuscript line 176.
And improved the relevant contents of the ethics statement, including "All animal
experiments(Approval number: XJIMM-20211006) were reviewed and approved by the
Ethics Committee of the Xinjiang Institute of Materia Medica. Animal welfare was
monitored daily by animal care staff. The approval document (Ethics Statements) has
been uploaded to other Item. Human participants and/or tissues were not involved in
this study."

4. Provided the original underlying images for Western blot analysis as Supporting
Information files at the end of manuscript, and the citations in the manuscript are
updated accordingly.

5. We checked the reference list of the manuscript to ensure that it was complete and
correct.

Thank you for considering our revised manuscript for publication. We look forward to
any additional comments or questions.

Yours sincerely,

Xu Hu

Reviewer #1: 

Dear editors and peer reviewers, 

First of all, it's a great honor to get your valuable suggestions. I attach great
importance to your review conclusion and will revise it according to your
suggestions to make the article more complete and rigorous. The following are my
answers to your Questions, hoping to get your approval.

Question 1. Why only male rats were chosen for the study?

Answer: Studies have shown that the physiological structure of female mammals is more
complex than that of male mammals, which is subject to certain uncertainties such as
pregnancy and estrous cycle, which may increase the experimental uncertainty.
Besides, this experiment involves indicators of inflammation. Female rats have
different sensitivities to inflammation. In conclusion, male rats were selected for
this study.

Question 2. Lot or CAT numbers for the kits and antibodies used throughout the study
were not mentioned, although it’s crucial for such a study.

Answer: Corresponding changes have been made in the revised manuscript line 162.

SDS-PAGE gel preparation kit (P0012A, Beyotime Biotechnology Company, China ),
lactate dehydrogenase (LDH) release assay kit (A020-2-2, Nanjing Jiancheng Institute
of Bioengineering, China), Nitric Oxide (NO) assay kit(A013-2-1, Nanjing Jiancheng
Institute of Bioengineering, China), Malondialdehyde (MDA) assay kit (A003-4-1,
Nanjing Jiancheng Institute of Bioengineering, China), IL-1β ELISA kit (E-EL-R0012c,
Elabscience Biotechnology, China), IL-6 ELISA kit (E-EL-R0015c, Elabscience
Biotechnology, China), TNF-α ELISA kit (E-EL-R2856c, Elabscience Biotechnology,
China), BCA protein quantification kit (23225, Thermo Fisher Pierce, US), primary
antibodies AKT(#4060, CST Company, US), p-AKT(#9272, CST Company, US), SRC(ab40660,
Abcam, US), β-actin (ab8226, Abcam, US), GAPDH antibody(TA-08, Zhongshan Jinqiao
Company, China), horseradish enzyme-labeled goat anti-mouse IgG (ZB-2305, Zhongshan
Jinqiao Company, China), horseradish enzyme-labeled goat anti-rabbit IgG (ZB-2301,
Zhongshan Jinqiao Company, China)

Question 3. After blood collection and centrifugation, how was the supernatant
stored? Also, how would the storage conditions affect LDH, MDA and NO levels?

Answer: After centrifugation, the supernatant was divided into several EP tubes and
stored in the refrigerator at -20℃. Before use, it should be melted at 4℃ and then
used. In order to ensure the accuracy and reliability of the determination results
of LDH, MDA and NO, samples should be kept at low temperature, sealed and neutral
conditions for determination as soon as possible. Our sample storage conditions can
meet the requirements.

Question 4. What kind of effect is being validated by administering TFDM before the
surgery not just after the surgery?

Answer: In this research, we use preventive medication. The purpose of preventive
administration is to observe the initial efficacy. We consider that if even
preventive administration can't see the efficacy, there is no need to conduct an
investigation on therapeutic administration.

Question 5. What was the success rate of establishing the CIRI rat model? As it’s
known such invasive procedure has a mortality rate. Were all the rats assigned from
the beginning alive throughout the rest of the study?

Answer: After the training of model establishment, before the experiment, the
modeling rate of CIRI rat model was about 70%. Because after the rat model is made,
it will cause certain death due to anesthesia, arterial rupture, cerebral hemorrhage
and other reasons, and the surviving rats are all alive before the subsequent blood
collection and execution.

Question 6. As stated in the study, TFDM shows promise as a therapeutic strategy for
CIRI. Would that indicate the use of TFDM as a prophylactic in general prior to the
condition or after the ischemic injury?

Answer: At present, TFDM can be considered as a candidate drug for preclinical
development. After completing the systematic preclinical research, we can enter the
clinical stage research, which needs to be tested on human body, so the drug must be
approved by the national drug supervision and administration department. In China,
research and development institutions of new drugs need to submit clinical
applications for new drugs to the National Medical Products Administration (NMPA),
and only after obtaining permission can they conduct human clinical trials. Clinical
research is divided into four stages: phase I clinical trial, phase II clinical
trial, phase III clinical trial and phase IV clinical research (post-marketing
monitoring of drugs). After completing the clinical trials in the above three
stages, the clinical data are statistically analyzed, which proves that the drug is
safe and effective, and at the same time, the drug quality is controllable, that is,
the three points of safety, effectiveness and controllability meet the requirements,
and the drug marketing license holder submits a new drug marketing application to
the drug supervision department. After obtaining the approval of the drug
supervision department, the drug can be marketed for doctors and patients to
choose.

Thank you for considering our revised manuscript for publication. We look forward to
any additional comments or questions.

Yours sincerely,

Xu Hu

Reviewer #2: 

Dear editors and peer reviewers,

First of all, it's a great honor to get your valuable suggestions. I attach great
importance to your review conclusion and will revise it according to your
suggestions to make the article more complete and rigorous. The following are my
answers to your Question, hoping to get your approval.

Question 1. Animal ethical Committee number should be mentioned in the
manuscript.

Answer: It has been revised in the revised manuscript line 176, and the approval
number is XJIMM-20211006.

Question 2. The grouping of animals and drug treatment pattern is not clear.

Answer: It has been revised in the revised manuscript line 179. After the rats are
bought back, they are kept in the animal house for 7 days to adapt to the
environment. After the rats adapt to the environment, the SD rats were randomly
divided into a sham operation group, a model group, and TFDM 30, 60 and 120 mg
groups, a total of 5 groups, 10 rats in each group.The rats of the three TFDM dose
groups were intragastrically administered with TFDM 30, 60 and 120 mg·kg-1 per day 7
d before surgery, respectively. The rats of the sham operation group and the model
group were intragastrically administered with corresponding doses of normal saline,
it lasted for 7 days, on the morning of the eighth day, the CIRI model was
established two hours after the administration was completed, 45min after the
operation caused cerebral ischemia, and 24 hours after reperfusion, it created a
cerebral ischemia-reperfusion injury model, and the samples were taken 24 hours
after operation.

Question 3. I note that there have been similar research works published in the past.
To address this issue, I recommend that you provide a clear explanation of how your
current research builds upon and/or differs from the existing literature.

Answer: Our research group comprehensively verified the prediction results based on
bioinformatics prediction data and combined with biological experiments to avoid
false positive or false negative results caused by relying entirely on
"bioinformatics". Combining computer virtual data with real pharmacological
experimental data to reveal the mechanism of TFDM in treating cerebral
ischemia-reperfusion injury comprehensively, accurately and efficiently is one of
the innovations of this topic. At the same time, we screened the active components
in TFDM by molecular docking with key targets. It provides a preliminary basis for
promoting the separation of chemical components of natural drugs and their
application in the treatment of brain diseases.

Question 4. Some of the references are not as per the journal guidelines

Answer: Corresponding changes have been made in the revised manuscript References
section.

Thank you for considering our revised manuscript for publication. We look forward to
any additional comments or questions.

Yours sincerely,

Xu Hu

Reviewer #3: 

Dear editors and peer reviewers,

First of all, it's a great honor to get your valuable suggestions. I attach great
importance to your review conclusion and will revise it according to your
suggestions to make the article more complete and rigorous. The following are my
answers to your Question, hoping to get your approval.

Question 1. In Abstract section line 18, authors should replace “Disease” with
“injury” as cerebral ischemia- reperfusion injury (CIRI) is usually caused by caused
by ischemic stroke seriously or restriction of blood supply. Line 28, 90, in vitro
should be written in Italic. Abstract is lack of key findings of the study.

Answer: Corresponding changes have been made in the revised manuscript Abstract
section. 

Some additional molecular docking results showed that acacetin-7-O (- 6''-acetyl)
-glucopyranoside, acacetin7-O-β-D-glucopyranoside, apigenin-7-O-β-D-galactoside
respectively had good affinity for SRC protein. acacetin-7-O (- 6''-acetyl)
-glucopyranoside,

acacetin-7-O-β-D-glucuronide, acacetin7-O-β-D-glucopyranoside had good affinity for
AKT1 protein, respectively. acacetin-7-O (- 6''-acetyl) -glucopyranoside is expected
to be used as a candidate monomer component for natural drugs for further
development.

Question 2. The ethical approval number should be added in methodology section as
authors mentioned the study has received approval from the Ethics Committee of the
Xinjiang Institute of Materia Medica.

Answer: It has been revised in the revised manuscript line 176, and the approval
number is XJIMM-20211006.

Question 3. Safety profile of total flavonoid extract of Dracocephalum Moldavica L.
should be addressed.

Answer: This research group investigated the safety of TFDM from the following two
aspects: 

1)To investigate the acute toxicity and death of TFDM after one-time administration
to mice and multiple times in a day. 

2)To investigate the toxic reaction of TFDM induced by gavage for consecutive 25
weeks, including the first symptom and severity, target organ of toxic reaction and
its recovery and development. 

Results 1) showed that TFDM given to mice by gavage with the maximum dose of
19.2g/kg.d (equivalent to 711 times of the proposed clinical dose) in a day did not
produce significant acute toxicity during the specified observation period. It
indicated that this preparation had no obvious acute toxicity and its medication was
safe. 

Results 2) Compared with the control group, no abnormal changes related to drug
administration were observed in rats' body weight, organ coefficient, hematology,
and blood biochemistry indexes. Histological examination also revealed no
pathological changes related to drug effects; It was considered that TFDM had no
obvious toxicity to rats when given by gavage within the specified dose and time
limit.

Question 4. Please add software specifications used in statistical analysis (Graph
prism). More robust & appropriate statistical analysis is also required.

Answer: Statistical analysis was performed with GraphPad Prism 8.0 software
Version8.0.2(263). All data were expressed as the mean ± standard deviation (SD).
The differences among multiple groups were evaluated using the one-way analysis of
variance (ANOVA). The difference between the means was considered statistically
significant at P < 0.05.

Question 5. Limitation of the study should be discussed in limitation section of the
manuscript.

Answer: Corresponding changes have been made in the revised manuscript line 413. 

One of the shortcomings of traditional Chinese medicine in treating diseases is its
slow onset, and the rapid onset of chemical drugs is its advantage over traditional
Chinese medicine. Therefore, the rational combination of traditional Chinese
medicine and chemical drugs can be considered at present, which provides a new
direction for anti-CIRI research.

In the future, we will conduct further research on the active components in TFDM that
have strong binding affinity with key targets.We believe that with the deepening of
the research on the pathogenesis of CIRI, the research on the mechanism of action,
treatment time window and pharmacokinetics of TFDM and its compound anti-CIRI is
becoming increasingly clear, and the current situation of CIRI treatment
difficulties will certainly make a breakthrough.

Question 6. Authors should describe the Future perspective and clinical significance
of the study.

Answer: Corresponding changes have been made in the revised manuscript line 354. 

Cerebral ischemia-reperfusion injury(CIRI) is a common and frequently-occurring
disease in the clinic. The mortality and disability rate are extremely high. It
seriously threatens human health and life, and the burden on society and families is
immeasurable. Therefore, scientists and clinicians need to make great efforts to
find potential pathogenic mechanisms, so as to take appropriate preventive measures
and successful treatment methods. CIRI involves a variety of mechanisms, including
free radical action, excitatory amino acids, cellular Ca2+ overload. To understand
the mechanism of CIRI and deepen the understanding of the process of cerebral
ischemic injury is conducive to the research and selection of medication. so as to
achieve the purpose of treatment. Natural medicines have a long history of treatment
for cerebral ischemia and have accumulated rich theoretical knowledge. Screening
active ingredients from natural medicines is also the current trend of research and
development. The whole world is promoting the separation of chemical components of
natural medicines and their application in the treatment of brain diseases. Our
research group has continuously conducted a series of studies on this issue, and
have made great progress. This study found that TFDM has unique advantages and
potential in the treatment of CIRI because of its diverse chemical components and
structures, and can intervene the pathological process of CIRI through multiple
targets and channels as a whole. In this paper, TFDM pharmacodynamics and its
mechanism of action are taken as the breakthrough point, in order to provide
theoretical basis for drug research and development of CIRI and scientific basis for
in-depth research on prevention and treatment of CIRI in traditional Chinese
medicine.

Question 7. Authors should add abbreviations list in the manuscript.

Answer: Corresponding changes have been made in the revised manuscript.

cerebral ischemia-reperfusion injury (CIRI) 

total flavonoid extract of Dracocephalum moldavica L. (TFDM) 

recombinant tissue plasminogen activator (rtPA) 

Traditional Chinese Medicine (TCM) 

Dracocephalum Moldavica L. (DML) 

effective parts of dracocephalum moldavica (EPDM) 

protein-protein interaction (PPI) 

Gene ontology (GO)

Kyoto Encyclopedia of Genes and Genomes (KEGG) 

molecular function (MF)

biological process (BP)

cellular components (CC)

horseradish peroxidase (HRP)

analysis of variance (ANOVA)

standard deviation (SD)

radio immuno precipitation assay (RIPA)

phenyl methyl sulfonyl fluoride (PMSF)

lactate dehydrogenase (LDH)

Malondialdehyde (MDA)

Thank you for considering our revised manuscript for publication. We look forward to
any additional comments or questions.

Yours sincerely,

Xu Hu

to Reviewers.docx
---

## [Decision Letter · Decision Letter 1]

12 Jul 2023

A network pharmacology approach to decipher the total flavonoid extract of
Dracocephalum Moldavica L. in the treatment of cerebral ischemia-
reperfusion injury

PONE-D-22-35325R1

Dear Dr. Hu,

We’re pleased to inform you that your manuscript has been judged scientifically
suitable for publication and will be formally accepted for publication once it meets
all outstanding technical requirements.

Kind regards,

Yong Sze Ong

Academic Editor

PLOS ONE

Additional Editor Comments (optional):

Reviewers' comments:

Reviewer's Responses to Questions

**Comments to the Author**

1. If the authors have adequately addressed your comments raised in a previous round
of review and you feel that this manuscript is now acceptable for publication, you
may indicate that here to bypass the “Comments to the Author” section, enter your
conflict of interest statement in the “Confidential to Editor” section, and submit
your "Accept" recommendation.

Reviewer #1: All comments have been addressed

Reviewer #3: All comments have been addressed

2. Is the manuscript technically sound, and do the data
support the conclusions?

Reviewer #1: Yes

Reviewer #3: Partly

3. Has the statistical analysis been performed
appropriately and rigorously? 

Reviewer #1: Yes

Reviewer #3: Yes

4. Have the authors made all data underlying the
findings in their manuscript fully available?

Reviewer #1: Yes

Reviewer #3: Yes

5. Is the manuscript presented in an intelligible
fashion and written in standard English?

Reviewer #1: Yes

Reviewer #3: Yes

6. Review Comments to the Author

Reviewer #1: I would like to thank the authors for addressing all the questions in
detail and considering the necessary changes.

Reviewer #3: (No Response)

7. PLOS authors have the option to publish the peer
review history of their article (what does this mean?). If published, this will
include your full peer review and any attached files.

If you choose “no”, your identity will remain anonymous but your review may still be
made public.

**Do you want your identity to be public for this peer review?** For
information about this choice, including consent withdrawal, please see our
Privacy Policy.

Reviewer #1: No

Reviewer #3: No

---

## [Editor Report · Acceptance letter]

17 Jul 2023

PONE-D-22-35325R1 

A network pharmacology approach to decipher the total flavonoid extract of
Dracocephalum Moldavica L. in the treatment of cerebral ischemia- reperfusion
injury  

Dear Dr. Hu:

I'm pleased to inform you that your manuscript has been deemed suitable for
publication in PLOS ONE. Congratulations! Your manuscript is now with our production
department. 

Kind regards, 

on behalf of

Dr. Yong Sze Ong 

Academic Editor

PLOS ONE